# Planning for Implicit Coordination using FOND

**Thorsten Engesser**
Department of Computer Science
University of Freiburg, Germany
engesser@cs.uni-freiburg.de

**Tim Miller**
Department of Computing and Information Systems
University of Melbourne, Australia
tmiller@unimelb.edu.au

## Abstract

Epistemic Planning can be used to achieve implicit coordination in cooperative multi-agent settings where knowledge and capabilities are distributed between the agents. In these scenarios, agents plan and act on their own without having to agree on a common plan or protocol beforehand. However, epistemic planning is undecidable in general. In this paper, we identify a decidable fragment of epistemic planning that allows for arbitrary initial state uncertainty and nondeterminism, but where actions can never increase the uncertainty of the agents. We show that in this fragment, planning with and without implicit coordination can be reduced to fully observable nondeterministic (FOND) planning and that it shares the same computational complexity. We also provide a small case study, modeling the problem of multi-agent path finding with destination uncertainty in FOND, to show that our compilation approach can be successfully applied in practice.

## Introduction

Epistemic planning has gained increasing interest in recent years (Baral *et al.* 2017). One of the main features of epistemic planning is the support of *knowledge goals*. For example, epistemic planning is well-suited to model problems in which information is to be confidentially passed between agents. The assumption is usually that there exists an explicit or implicit model about the distributed knowledge of the agents, as well as actions which can change the models.

However, recent work has shown that epistemic planning can also be used to achieve *implicit coordination* in a setting where multiple agents plan and act for themselves towards a cooperative goal (Engesser *et al.* 2017). The idea is that the explicit modeling of the agents' knowledge can be exploited as a means to enforce coordination via *perspective taking*. In particular, by putting themselves into the shoes of the others, agents can account for possible contributions of other agents in their own plans. Bolander *et al.* (2018) showed under which conditions such plans are guaranteed to be successful.

This problem of planning for implicit coordination was originally formalized as a variant of contingent planning in the space of epistemic states (i.e., Kripke models), with actions represented by the action models from Dynamic Epistemic Logic (van Ditmarsch *et al.* 2007). The formalization is very similar to the one of Bolander and Andersen (2011), which produces action sequences that can be interpreted as centralized plans. Bolander and Andersen have shown that this type of epistemic planning is undecidable in general. However, some decidable fragments have been identified that rely on restricting the structure of action models and the form of allowed preconditions (Aucher and Bolander 2013; Bolander *et al.* 2015; Charrier *et al.* 2016). On the practical side, Kominis and Geffner (2015) and Muise *et al.* (2015) have identified fragments of epistemic planning that can be solved by compilation to classical planning.

In this paper, we define a decidable fragment that captures contingent epistemic planning and that can be compiled to fully-observable nondeterministic (FOND) planning. Our fragment generalizes the fragment of Kominis and Geffner. We then show how our compilation can be extended to capture planning for implicit coordination. The key insight is that we can use nondeterminism to simulate perspective taking and thus account for the imperfect knowledge of the agents.

## Theoretical Background

We will first recapitulate the DEL planning framework using the conventions of Bolander and Andersen (2011), but including conditional effects in the style of van Benthem *et al.* (2006). We will then review strong fully-observable nondeterministic planning (Cimatti *et al.* 2003; Ghallab *et al.* 2004) as well as planning for implicit coordination (Engesser *et al.* 2017; Bolander *et al.* 2018).

### The DEL Planning Framework

For a fixed set of agents $\mathcal{A}$ and a fixed set of atomic propositions $P$, the epistemic language $\mathcal{L}_{KC}$ is given by the BNF

$$\phi ::= p \mid \neg\phi \mid \phi \wedge \phi \mid K_i\phi \mid C\phi, \text{ where } p \in P \text{ and } i \in \mathcal{A}.$$

We read $K_i\phi$ as "agent $i$ knows $\phi$" and $C\phi$ as "it is common knowledge between all agents that $\phi$". The additional connectives $\vee, \leftarrow, \rightarrow, \leftrightarrow$ can be defined as abbreviations, analogously to their definition in propositional logic.

We evaluate such formulas over *epistemic models*. An epistemic model is a tuple $\mathcal{M} = \langle W, (R_i)_{i \in \mathcal{A}}, V \rangle$, where

$W$ is a non-empty, finite set of worlds (the *domain* of $\mathcal{M}$), $R_i \subseteq W \times W$ is an equivalence relation for each agent $i \in \mathcal{A}$ (the *indistinguishability relation* of $i$), and with $V : P \to 2^W$ (the *valuation function*). We write $R^*$ for the transitive closure of $\bigcup_{i \in \mathcal{A}} R_i$. The truth of a formula $\phi \in \mathcal{L}_{KC}$ in a world $w$ of a model $\mathcal{M}$ is then given as follows, where the propositional cases are standard and hence left out:

$$\begin{array}{lll} \mathcal{M}, w \models p & \text{iff} & w \in V(p) \\ \mathcal{M}, w \models K_i\phi & \text{iff} & \mathcal{M}, w' \models \phi \text{ for all } wR_iw' \\ \mathcal{M}, w \models C\phi & \text{iff} & \mathcal{M}, w' \models \phi \text{ for all } wR^*w' \end{array}$$

We depict epistemic models as graphs where nodes correspond to the worlds in the model and are additionally labeled with the atomic propositions that are true in that particular world. The indistinguishability relations are given as labeled edges between the worlds. For readability, we will omit reflexive edges as well as edges that are implied by transitivity. Consider the following epistemic model:

$$\mathcal{M}_0 = \underset{w_1 : p}{\bullet} \overset{1,2}{\rule{2cm}{0.4pt}} \underset{w_2 :}{\bullet}$$

In our example, both agents 1 and 2 do not know whether or not $p$ is true (which is the case in $w_1$) or false (which is the case in $w_2$). Also, it is common knowledge between the two agents that they do not know. We will now define example actions for agent 1, first to sense the value of $p$ and then to announce it to agent 2.

To define actions, we use *event models*. These can change the facts about the world as well as the knowledge of the agents. Analogous to epistemic models, an event model is a tuple $\mathcal{E} = \langle E, (Q_i)_{i \in \mathcal{A}}, \text{pre}, \text{eff} \rangle$, where $E$ is a non-empty, finite set of events (the *domain* of $\mathcal{E}$) and $R_i \subseteq W \times W$ is an equivalence relation for each agent $i \in \mathcal{A}$ (the *indistinguishability relation* of $i$). Instead of a valuation function, we have two functions $\text{pre} : E \to \mathcal{L}_{KC}$ and $\text{eff} : E \to (P \to \mathcal{L}_{KC})$, assigning a *precondition* and conditional *effects* to each event.

We depict event models analogously to epistemic models with the difference that nodes now correspond to events, which are additionally labeled with their respective preconditions and effects. Consider the following event model:

$$\mathcal{E}_{\text{sense}} = \underset{e_1 : \langle p, \{p \mapsto p\} \rangle}{\bullet} \overset{2}{\rule{2cm}{0.4pt}} \underset{e_2 : \langle \neg p, \{p \mapsto p\} \rangle}{\bullet}$$

An event model *updates* an epistemic model by pairing up every world with every applicable event (i.e., of which the precondition is satisfied). Two updated worlds are indistinguishable for an agent if both the original worlds and the events are indistinguishable for that agent. Furthermore, a proposition is true in an updated world if and only if the event's conditional effect concerning that proposition evaluates to true in the original world.

For example, $\mathcal{E}_{\text{sense}}$ consists of two events with preconditions $p$ and $\neg p$. For both events, the effect is $\{p \mapsto p\}$ meaning $p$ will be true if $p$ was true before (from now on, we will omit these trivial effects that preserve the value of an atomic proposition in our depiction of event models). Since

the events are distinguishable for agent 1, the agent will, after the execution of the action, be able to distinguish worlds in which $p$ is true from worlds in which $p$ is false.

Formally, we define the *product update* $\mathcal{M} \otimes \mathcal{E}$ of model $\mathcal{M} = \langle W, (R_i)_{i \in \mathcal{A}}, V \rangle$ with respect to an event model $\mathcal{E} = \langle E, (Q_i)_{i \in \mathcal{A}}, \text{pre}, \text{eff} \rangle$ as model $\langle W', (R'_i)_{i \in \mathcal{A}}, V' \rangle$ where

- $W' = \{(w, e) \in W \times E \mid \mathcal{M}, w \models \text{pre}(e)\}$,
- $R'_i = \{((w, e), (w', e')) \in W' \times W' \mid wR_iw', eQ_ie'\}$,
- $V'(p) = \{(w, e) \in W' \mid \mathcal{M}, w \models \text{eff}(e, p)\}$.

In particular, if we apply $\mathcal{E}_{\text{sense}}$ in $\mathcal{M}_0$, we obtain the following epistemic model:

$$\mathcal{M}_0 \otimes \mathcal{E}_{\text{sense}} = \underset{(w_1, e_1) : p}{\bullet} \overset{2}{\rule{2cm}{0.4pt}} \underset{(w_2, e_2) :}{\bullet}$$

As intended, agent 1 knows now whether or not $p$ is true. Note that additionally agent 2 is aware of this. The event model $\mathcal{E}_{\text{sense}}$ represents *semi-private sensing*, meaning that even though the result of the sensing will only be known to agent 1, agent 2 will know that the sensing has taken place.

For planning, we usually consider pointed models $(\mathcal{M}, w)$, i.e., where one world $w$ from the domain of $\mathcal{M}$ is designated as the actual world. In contrast, we model *epistemic actions* as multi-pointed event models $(\mathcal{E}, E_d)$ where $E_d$ is a subset of the domain of $\mathcal{E}$. This is necessary, since sometimes we want the events to be deliberately chosen by the acting agents and sometimes by the environment. E.g., our semi-private sensing action should be defined as $(\mathcal{E}_{sense}, \{e_1, e_2\})$. Since both events are designated, it can be applied regardless of whether $p$ is true or false. Applied in $(\mathcal{M}_0, w_1)$, the action results in the pointed model $(\mathcal{M}_0 \otimes \mathcal{E}_{\text{sense}}, (w_1, e_1))$ and applied in $(\mathcal{M}_0, w_2)$ it results in $(\mathcal{M}_0 \otimes \mathcal{E}_{\text{sense}}, (w_2, e_2))$. The similar action $(\mathcal{E}_{sense}, \{e_1\})$ is only applicable in the case where $p$ is true. It can, e.g., be used to model the action of a third agent semi-privately informing agent 1 that $p$ is true.

Formally, an epistemic action $(\mathcal{E}, E_d)$ is applicable in $(\mathcal{M}, w)$ if there is an applicable event $e \in E_d$, meaning that $\mathcal{M}, w \models \text{pre}(e)$. The application of $(\mathcal{E}, E_d)$ in $(\mathcal{M}, w)$ then nondeterministically leads to a pointed model $(\mathcal{M} \otimes \mathcal{E}(w, e))$ such that $\mathcal{M}, w \models \text{pre}(e)$.

Note that any *epistemic state* represented by a pointed model $(\mathcal{M}, w)$, has infinitely many epistemically equivalent representations (i.e., other pointed models that satisfy the exact same set of formulas). It is a central theorem of modal logic that finite models are epistemically equivalent if and only if they are bisimilar. In the following, when using pointed models as states in a transition system, we think of them as representatives of their whole equivalence class. I.e., we consider two epistemic states $(\mathcal{M}, w)$ and $(\mathcal{M}', w')$ as identical if they are epistemically equivalent. And we say two epistemic states $(\mathcal{M}, w)$ and $(\mathcal{M}', w')$ are indistinguishable for an agent $i$ if there is a world $w''$ in $\mathcal{M}$ that is indistinguishable to $w$ for agent $i$ such that $(\mathcal{M}, w'')$ and $(\mathcal{M}', w')$ are identical. An initial epistemic state together with a set of epistemic actions thus induces a nondeterministic transition system where all states are epistemically different from each other.

## FOND Planning

Our definition of FOND planning loosely follows the conventions of Ghallab *et al.* (2004). In particular, our actions consist of one common precondition and a set of possible effects, from which one will always be chosen nondeterministically. However, since we want to start out with a formalization that is as close as possible to our DEL formalism, we allow arbitrary propositional formulas as action preconditions and goals. We also use *conditional effects* which we restrict to *effect normal form*, which is a special case of Rintanens unary conditionality normal form (Rintanen 2003).

We define a FOND planning task as a tuple $\langle \mathcal{F}, \mathcal{I}, \gamma, Act \rangle$ where $\mathcal{F}$ is a set of fluents (atomic propositions), $\mathcal{I} \subseteq \mathcal{F}$ is the initial state, $\gamma$ is a propositional goal formula over $\mathcal{F}$ and $Act$ is a set of actions. Each action $a = \langle \mathrm{pre}_a, \mathrm{effs}_a \rangle \in Act$ consists of a propositional formula $\mathrm{pre}_a$ over $\mathcal{F}$ (the precondition) and a set $\mathrm{effs}_a$ (the conditional effects). Each conditional effect $e \in \mathrm{effs}_a$ is of the form $\bigwedge_{f \in \mathcal{F}} (\chi_f^e \rhd f) \wedge (\chi_{\neg f}^e \rhd \neg f)$, where $\chi_f^e$ and $\chi_{\neg f}^e$ are mutually inconsistent propositional formulas over $\mathcal{F}$ (i.e., their conjunction is unsatisfiable). They can be interpreted as "effect $e$ makes $f$ true under the condition $\chi_f^e$ and false under the condition $\chi_{\neg f}^e$". Such a FOND task induces a finite transition system starting with the initial state $I$ and connecting two states $\mathcal{S}$ and $\mathcal{S}'$ via action $a$ iff $S \models \mathrm{pre}_a$ and there is an effect $e \in \mathrm{effs}_a$ such that the conditional effects in $e$ transform $\mathcal{S}$ to $\mathcal{S}'$.

This gives us a trivial compilation from FOND to DEL. I.e., we compile the initial state into an epistemic state with exactly one world $w_0$ where $V(p) = \{w_0\}$ iff $p \in \mathcal{I}$, or $\emptyset$ otherwise. And for each action $a \in Act$, we construct an epistemic action with one event for each nondeterministic effect $e \in \mathrm{effs}_a$, with precondition $\mathrm{pre}_a$ and effect $\{f \mapsto \chi_f^e \vee (f \wedge \neg \chi_{\neg f}^e) \mid f \in \mathcal{F}\}$. All events are designated and pairwise distinguishable for all agents. The transition system that we get from our compiled DEL state and actions is isomorphic to the FOND transition system and identified states share the same propositional valuation.

One solution to FOND planning tasks are *strong plans*. These are partial functions $\pi$ from states to actions which satisfy the following properties (Cimatti *et al.* 2003):

- For every state $s$ that is reachable via $\pi$ from $I$, there is some state $s'$ that is reachable from $s$ via $\pi$, s.t. $s' \models \gamma$.

- There are no cycles, i.e. states $s$ and $s'$ such that $s'$ is reachable via $\pi$ from $s$ and $s'$ is reachable via $\pi$ from $s$.

Since the transition system is finite, following a strong policy always leads to a goal state in finitely many steps. It seems reasonable to assume that the concept of strong policies is also useful for contingent planning over epistemic states.

## Implicit Coordination in DEL

We define an epistemic planning task as a tuple $\langle s_0, A, \omega, \gamma \rangle$ where $s_0$ is an epistemic state (the *initial state*), A is a finite set of epistemic actions (the *action library*), $\omega : A \to \mathcal{A}$ is a function mapping each action to its owner (the *owner function*), and $\gamma \in \mathcal{L}_{KC}$ is the *goal formula*.

E.g., consider the planning task with $s_0 = (\mathcal{M}_0, w_1)$, $A = \{\mathrm{sense}, \mathrm{ann}_p, \mathrm{ann}_{\neg p}\}$ with $\mathrm{sense} = (\mathcal{E}_{\mathrm{sense}}, \{e_1, e_2\})$, $\mathrm{ann}_p = (\mathcal{E}_{\mathrm{ann}_p}, e_1)$ and $\mathrm{ann}_{\neg p} = (\mathcal{E}_{\mathrm{ann}_{\neg p}}, e_1)$. The actions $\mathrm{ann}_p$ and $\mathrm{ann}_{\neg p}$ are public announcement actions for agent 1, announcing that $p$ is true, or respectively false. That is, the event models $\mathcal{E}_{\mathrm{ann}_{\neg p}}$ and $\mathcal{E}_{\mathrm{ann}_p}$ are given as follows:

$$\mathcal{E}_{\mathrm{ann}_p} = \overset{\bullet}{e_1 : \langle p, \emptyset \rangle} \qquad \mathcal{E}_{\mathrm{ann}_{\neg p}} = \overset{\bullet}{e_1 : \langle \neg p, \emptyset \rangle}$$

We assume that all actions are owned by agent 1, i.e., $\omega = \{\mathrm{sense} \mapsto 1, \mathrm{ann}_p \mapsto 1, \mathrm{ann}_{\neg p} \mapsto 1\}$. The goal is for agent 2 to know whether or not $p$ is true, i.e., $\gamma = K_2 p \vee K_2 \neg p$.

A strong policy in the sense of Cimatti *et al.* (2003) would be to just apply the action $\mathrm{ann}_p$ in $s_0$. This is because the action is applicable in $(\mathcal{M}_0, w_1)$ and its application would lead to a successor state consisting of only one world $(w_1, e_1)$ in which $p$ (and $K_2 p$) is true. We argue that from the perspective of the agents (who initially do not know whether $p$ is true or false), this is not a reasonable solution. If we want agent 1 to be able to come up with the plan for himself, we must consider his incomplete knowledge about the situation. Intuitively, a good plan for agent 1 is to first apply the sensing action and then, depending on the sensing result, apply the action $\mathrm{ann}_p$ or $\mathrm{ann}_q$. This plan works for both states $(\mathcal{M}, w_1)$ and $(\mathcal{M}, w_2)$, which agent 1 considers possible.

To capture this, we have to require uniform policies. A *uniform* policy is a partial function $\pi$ from epistemic states to sets of epistemic actions, satisfying the following constraints:

- Applicability: for each state $s$, and action $a \in \pi(s)$, the action $a$ has to be applicable in state $s$.

- Uniformity: for each state $s$, and action $a \in \pi(s)$, and states $s'$ that are indistinguishable to $s$ for the owner $\omega(a)$ of the action, also $a \in \pi(s')$.

This definition ensures that the agents can always infer from their own knowledge whether or not and how the policy wants them to act. This also implies that an action is only applicable by an agent, if the agent knows that the action is applicable. Note that because of the uniformity constraint, it is necessary to allow policies to assign multiple actions per state. E.g., sometimes we want a policy to assign an action $a$ of agent 1 to some state $s$ and an action $b$ of agent 2 to some state $s'$. Then by uniformity, if there is a state $s''$ that is indistinguishable to $s$ for agent 1 and to $s'$ for agent 2, we have to assign both $a$ and $b$ to $s''$.

We then say a uniform policy is *subjectively strong*, if it satisfies the exact properties of strong plans, but based on *subjective reachability*: A state $s'$ is a subjective successor of $s$ given an action $a$ if there is a successor state of $s$ and $a$ that is indistinguishable to $s'$ for agent $\omega(a)$. I.e., in our example, the subjective successors of $(\mathcal{M}_0, w_1)$ and $(\mathcal{E}_{\mathrm{sense}}, \{e_1, e_2\})$ are exactly the states $(\mathcal{M}_0 \otimes \mathcal{E}_{\mathrm{sense}}, (w_1, e_1))$ and $(\mathcal{M}_0 \otimes \mathcal{E}_{\mathrm{sense}}, (w_2, e_2))$. A state $s'$ is then subjectively reachable from $s$ if either $s'$ is identical to $s$ or $s'$ is subjectively reachable from a subjective successor of $s$.

In particular, a policy $\pi$ that is subjectively strong for an epistemic planning task $\langle s_0, A, \omega, \gamma \rangle$ guarantees for each subjectively reachable state $s$ and action $a \in \pi(s)$, that $\pi$

is also subjectively strong for $\langle s, A, \omega, \gamma \rangle$, as well as for all planning tasks $\langle s', A, \omega, \gamma \rangle$ with an initial state $s'$ that is indistinguishable to $s$ for $\omega(a)$.

## A Decidable Fragment of DEL Planning

A straight-forward way to obtain a decidable fragment of DEL planning is to ensure that the induced transition system is finite. This can be done by restricting the action set in such a way that the application of a single action can never lead to a state where the number of worlds is greater than in the state in which the action was applied. We achieve this by requiring our event models to be partitioned into disjoint connected components with mutually inconsistent preconditions. This allows us to think of each of the components as a single nondeterministic effect. Consider the following event model:

$$\mathcal{E}_{pp} = \boxed{\begin{array}{cc} \bullet \xrightarrow{\;\;2\;\;} \bullet \\ e_1 : \langle p, \emptyset \rangle \quad e_2 : \langle \neg p, \emptyset \rangle \end{array}} \boxed{\begin{array}{c} \bullet \\ e_3 : \langle p, \emptyset \rangle \end{array}}$$

For example, the action $(\mathcal{E}_{pp}, \{e_1, e_3\})$ could model an agent 3 trying to semi-privately announce $p$ to agent 1. However, there is the possibility that the confidentiality of the announcement is compromised and $p$ is thus effectively publicly announced. If we apply this action in $(\mathcal{M}_0, w_1)$, it results either in $(\mathcal{M}_0 \otimes \mathcal{E}_{pp}, (w_1, e_1))$ or $(\mathcal{M}_0 \otimes \mathcal{E}_{pp}, (w_1, e_3))$ where

$$\mathcal{M}_0 \otimes \mathcal{E}_{pp} = \boxed{\begin{array}{cc} \bullet \xrightarrow{\;\;2\;\;} \bullet \\ (w_1, e_1) : p \quad (w_2, e_2) : \end{array}} \boxed{\begin{array}{c} \bullet \\ (w_1, e_3) : p \end{array}}$$

Formally, for actions $(\langle E, (Q_i), \text{pre}, \text{eff} \rangle, E_d)$ from our action set, we require that the domain $E$ can be partitioned into disjoint subsets $E_1, \ldots, E_k$ such that (1) for each pair of events $e, e' \in E_j$ from the same component $j \in \{1, \ldots, k\}$, the preconditions $\text{pre}(e)$ and $\text{pre}(e')$ are mutually inconsistent, and (2) two events $e, e' \in E$ are only allowed to be indistinguishable for an arbitrary agent $i \in \mathcal{A}$, i.e. $eQ_ie'$, if they belong to the same component, i.e if there exists a $j \in \{1, \ldots k\}$ such that $e, e' \in E_j$.

We can see that if we apply such an action to an arbitrary epistemic state, due to condition (1), each world will be paired up by maximally one of the events of each component. Furthermore, due to condition (2), two worlds can only be distinguishable for any agent if the events they were generated with are from the same component. Thus the state resulting from an action application will consist of at most $k$ connected components which consist each of less or equally many worlds than the original state. Since we can throw away all components that do not contain the updated designated world, we obtain a state that can be represented by less than or equally many worlds as the original state.

Our fragment is a generalization of the fragment introduced by Kominis and Geffner (2015), which allows exactly those actions that can be described with only mutually inconsistent preconditions (even between events from different components) and where all actions are thus deterministic.

## Compilation to FOND

In the following we will show how to generate a FOND planning task $\langle \mathcal{F}, \mathcal{I}, \gamma^*, Act \rangle$, given an epistemic planning task $\langle s_0, A, \omega, \gamma \rangle$ from our fragment.

### Compilation of Epistemic States

We use the approach of Kominis and Geffner (2015) to represent epistemic states as classical states. The idea is that we generate fluents directly from the worlds and indistinguishability relation of the initial state $s_0$, such that we can use them to encode the valuation functions and indistinguishability relations of arbitrary states reachable from $s_0$.

Given the initial state $s_0 = (\langle W, (R_i), V \rangle, w_0)$, we introduce a fluent $p^w \in \mathcal{F}$ (read: "$p$ is true in world $w$") for each proposition $p \in P$ and world $w \in W$. Similarly, we introduce a fluent $D_i^{\{w_1, w_2\}}$ (read: "$w_1$ is distinguishable to $w_2$ for agent $i$") for each agent $i \in \mathcal{A}$ and worlds $w_1, w_2 \in W$ with $w_1 R_i w_2$. Finally, for each world $w \in W$, we introduce the fluent $w^* \in \mathcal{F}$ (read: "$w$ is the designated world").

A propositional state $\mathcal{S} \subseteq \mathcal{F}$ then represents an epistemic state $(\langle W, (R_i'), V' \rangle, w)$ where (1) $w \in V'(p)$ iff $p^w \in \mathcal{S}$, (2) $w_1 R_i' w_2$ iff $D_i^{\{w_1, w_2\}} \notin \mathcal{S}$, and (3) $w = w'$ iff $w'^* \in \mathcal{S}$.

For example, if $s_0 = (\mathcal{M}_0, w_1)$, we will generate the set of fluents $\mathcal{F} = \{p^{w_1}, p^{w_2}, D_1^{\{w_1, w_2\}}, D_2^{\{w_1, w_2\}}, w_1^*, w_2^*\}$. The initial state $s_0$ will then be $\mathcal{I} = \{p^{w_1}, w_1^*\}$.

### Compilation of Epistemic Formulas

To check whether a propositional formula $\phi$ is true in world $w$ of an epistemic state that is represented by a classical state $\mathcal{S} \subseteq \mathcal{F}$ is simple. We replace the occurrences of each proposition $p$ in $\phi$ by $p^w$ and check the resulting formula in $\mathcal{S}$.

Checking formulas with knowledge operators is slightly more complicated. Kominis and Geffner (2015) use *axioms* to compile away all knowledge subformulas into *derived variables*, the values of which can be inferred in polynomial time.

We will simply assume that all of this is given and that we can thus compile each epistemic formula to a formula $\phi^w$ that evaluates to true in a classical state representing an epistemic state $(\mathcal{M}, w_0)$ iff $\mathcal{M}, w \models \phi$.

For evaluating a formula directly in the designated world of a state (e.g., the goal formula), we use $\phi^*$, which we define as $(\bigvee_{w \in W} w^*) \wedge \bigwedge_{w \in W} (w^* \rightarrow \phi^w)$.

### Compilation of Epistemic Actions

We now show how an action $a = (\langle E, (Q_i), \text{pre}, \text{eff} \rangle, E_d)$ that can be partitioned into distinct components $E_1, \ldots, E_k$ accordingly to our fragment, can be compiled into a FOND action $\langle \text{pre}_a, \text{effs}_a \rangle$.

We know that an action is applicable in a state $(\mathcal{M}, w)$ if there is some event $e \in E_d$ such that $\mathcal{M}, w \models \text{pre}(e)$. We directly translate this to $\text{pre}_a = \bigvee_{e \in E_d} \text{pre}(e)^*$. We can then translate each of the components of our event model into a different nondeterministic effect, i.e., we get $\text{effs}_a = \{\text{eff}_j \mid j = 1, ..., n\}$. These nondeterministic effects can make propositions true or false, as well as make

worlds distinguishable or completely inaccessible. We construct each nondeterministic effect $\text{eff}_j$ as follows:

$$\text{eff}_j = \text{eff}_j^{P+} \wedge \text{eff}_j^{P-} \wedge \text{eff}_j^{D+} \wedge \text{eff}_j^{\times} \wedge \text{eff}_j^{\times\times}$$

First, each fluent $p^w$ is made true or false accordingly to the effects of the event $e \in E_j$ that is applied in $w$.

$$\text{eff}_j^{P+} = \bigwedge_{\substack{w \in W \\ p \in P}} (\vee_{e \in E_j} (\text{pre}(e)^w \wedge \text{eff}(e, p)^w) \rhd p^w)$$

$$\text{eff}_j^{P-} = \bigwedge_{\substack{w \in W \\ p \in P}} (\vee_{e \in E_j} (\text{pre}(e)^w \wedge \neg \text{eff}(e, p)^w) \rhd \neg p^w)$$

Two worlds $w$ and $w'$ become distinguishable if the events $e$ and $e'$ they were updated with are distinguishable:

$$\text{eff}_j^{D+} = \bigwedge_{\substack{w, w' \in W \\ i \in \mathcal{A}, w \neq w'}} \left( \bigvee_{\substack{e, e' \in E_j \\ \neg e Q_i e'}} (\text{pre}(e)^w \wedge \text{pre}(e')^{w'}) \rhd D_i^{\{w, w'\}} \right)$$

If in some world $w$, none of the events from $E_j$ are applicable, the world should not have a successor. We simulate this by making $w$ distinguishable from all other worlds.

$$\text{eff}_j^{\times} = \bigwedge_{\substack{w, w' \in W \\ w \neq w'}} \left( \wedge_{e \in E_j} \neg \text{pre}(e)^w \right) \rhd D_i^{\{w, w'\}}$$

If for the designated world $w$, there is no applicable event in $E_j$, there should not even be a corresponding successor state. We model this by completely removing the designation $w^*$. Thus, while the effect is still applicable, it leads to a state where all formulas $\phi^*$ evaluate to false and therefore no actions are applicable and the goal is not satisfied.

$$\text{eff}_j^{\times\times} = \left( \wedge_{e \in E_j} \neg \text{pre}(e)^w \right) \rhd \neg w^*$$

### Compilation of Policies

Our compilation guarantees that the nondeterministic outcomes of an action that is applied in a propositional state corresponds exactly to the nondeterministic outcomes of the original epistemic action applied to the original epistemic state. Thus any strong policy for an epistemic planning task automatically corresponds to a strong policy in its FOND compilation. I.e., we can start in the initial state of the FOND compilation and extract a policy by successively applying the actions assigned by the original policy.

For the other direction, we can proceed similarly. However, we have to be careful about the fact that the policy can can contain multiple propositional states representing the same epistemic state. E.g., consider the states $\{p^{w_1}, w_1^*\}$ and $\{p^{w_2}, w_2^*\}$. Having equivalent states in a policy is unproblematic, as long as one is never reachable from the other. To obtain a strong policy for the original problem, we can apply the same policy extraction procedure from above but ignore each state if we have seen an equivalent state before.

If the policy in our FOND compilation contains equivalent states such that one is reachable from the other, the extraction gets more difficult, as we have to take care of not introducing cycles into our policy. Fortunately, it is easy to argue that if there is no strong policy in the FOND compilation which doesn't include equivalent states that are reachable from each other, there will also be no strong policy that

includes them. This is because the transition system looks exactly the same from these states and we do not gain anything from getting from one of the states to the other. This means that if there is a strong policy for the FOND compilation of an epistemic planning task, there also has to exist a strong policy that does not contain epistemically equivalent reachable states. Moreover, if the strong policy in the FOND compilation is optimal (i.e., its tree representation has minimal depth), it is clear that the policy cannot contain equivalent states that are reachable from each other. We thus obtain the following theorem.

**Theorem 1.** *Let $\Pi$ be an epistemic planning task from our fragment. Then there exists a strong policy for $\Pi$ if and only if there exists a strong policy for the FOND compilation of $\Pi$. Any optimal strong policy for $\Pi$ directly corresponds to an optimal strong policy for its compilation and vice versa.*

The following theorem follows, given the EXPTIME-completeness of the plan existence problem for strong planning in FOND (Rintanen 2004).

**Theorem 2.** *In our epistemic planning fragment, the problem of deciding whether there exists a strong policy for a given planning task is EXPTIME-complete.*

## Planning for Implicit Coordination

As explained in our section about planning for implicit coordination, strong policies are not suitable if we want the agents to coordinate implicitly. In this section, we show how to use FOND planning to find subjectively strong plans for epistemic planning tasks from our fragment.

### The Compilation

We use the same compilation of states and formulas as before. However, we slightly modify the compilation of actions. The idea is to split each action from the action set into two: One auxiliary action for *choosing* the action that we want to apply in a state and one action that actually *applies* the effects of the previously selected action. In each choice action, we additionally simulate a *perspective shift*: We change the designated world nondeterministically to any of the worlds that are indistinguishable for the owner agent of the action. Thus, subjective successors of an action in the original problem are now objective successors.

This means that any strong policy in the compilation will correspond to a subjectively strong policy in the original problem. We can extract such a policy by taking all the apply-actions from our policy and and assigning the corresponding actions to the corresponding states in the original planning task.

**Theorem 3.** *In our epistemic planning fragment, the problem of deciding whether there exists a subjectively strong policy is EXPTIME-complete.*

### Example: MAPF/DU

We demonstrate our approach by modeling an instance of *multi-agent path finding with destination uncertainty*. The problem was first described by Bolander *et al.* (2018) and

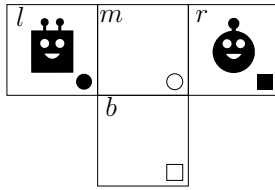

Figure 1: A MAPF/DU instance.

more thoroughly analyzed by Nebel *et al.* (2019). It is a generalization of the multi-agent path finding problem, relaxing the assumption that the agents' goals are commonly known. Instead, we assume that there are pairwise disjoint sets of plausible goal candidates for each agent, which are commonly known. Also, each agent can identify its own goal. As final action, each agent is allowed to announce that he has arrived at its true destination. The joint goal for the agents is that each agent is at his own true goal. Nebel *et al.* showed that the plan existence problem is PSPACE-complete. The naive algorithm they proposed has a runtime complexity of $O(n^{a^2+a})$ where n is the number of graph vertices and a is the number of agents.

Figure 1 shows an example of a MAPF/DU instance with two agents. The goal candidates of the square agent are $r$ and $b$, and the goal candidates of the circle agent are $l$ and $m$. One subjectively strong policy is for the square agent to first go to $b$ and to let the circle agent move to $l$, independently of the actual destinations of the agents. Then, the square agent goes to its true destination (which, depending on the designated world, will be either $r$ or $b$) and announces success there. Afterwards, the circle agent can go to his true destination (which will be either $l$ or $m$). Note that after the initial movements of the square agent, the policy has to consider all 4 possible goal combinations. This is because the square agent does not know the actual goal of the circle agent and the circle agent will not know the actual goal of the square agent.

We now show how this problem can be modeled in PDDL (McDermott 1998). We will use the types `agt` for agents, `pos` for positions, and `wld` for worlds. We introduce fluents `(at ?a ?p)` to denote that agent `?a` is at position `?p`, `(adj ?p ?q)` to denote that an agent can step from position `?p` to position `?q`, and `(announced ?a)` to denote that the agent `?a` has already announced success and will not move any longer. Furthermore, we use `(goal ?w ?a ?p)` to denote that the actual goal of agent `?a` in world `?w` is position `?p`.

To denote indistinguishability of two worlds `?w1` and `?w2` for agent `?a`, we use the fluent `(ind ?a ?w1 ?w2)`. We mark the designated world `?w` using the fluent `(des ?w)`.

Finally, we use the predicates `(next-choose)`, `(next-move ?a ?p1 ?p2)` and `(next-announce ?a)` to enforce the alternation of auxiliary perspective-shifting actions and actual actions.

We now show how to split up the movement actions into the actions `choose-move` and `move`. The action `(choose-move ?a ?w ?p ?q)` simulates a perspec-

tive shift to agent `?a` by nondeterministically switching to an arbitrary world that is indistinguishable from the designated world for agent `?a`. Furthermore, by setting the fluent `(next-move ?a ?p ?q)` to true, it enforces a movement action for agent `?a` from `?p` to `?q` in the successor state.

```
(:action choose-move
   :parameters (?a - agt ?w - wld ?p ?q - pos)
   :precondition (and (des ?w) (next-choose))
   :effect (and
     (not (next-choose))
     (next-move ?a ?p ?q)
     (oneof  ; simulate perspective shift
 (when (and (ind ?a ?w w1) (not (= ?w w1)))
       (and (not (des ?w)) (des w1)))
 (when (and (ind ?a ?w w2) (not (= ?w w2)))
       (and (not (des ?w)) (des w2)))
 ; ...
```

Unfortunately, we have to enumerate all possible worlds to simulate the perspective shift. This forces us to include the worlds as constants into the domain definition. It would be more convenient if we had a dedicated construct in PDDL to automatically generate nondeterministic effects, e.g., by explicitly quantifying over objects (in our case, worlds).

The `move` action, which has to be applied afterwards, performs the actual change of the agent's position. This action also contains the actual precondition for movement actions: the field to move to has to be adjacent and empty. Also, the action prescribes the next action to be again a `choose` action by setting the fluent `next-choose` to true.

```
(:action move
   :parameters (?a - agt ?w - wld ?p ?q - pos)
   :precondition (and
     (des ?w) (next-move ?a ?p ?q)
     (at ?a ?p) (adj ?p ?q)
     (not (announced ?a))
     (not (exists (?a2 - agt) (at ?a2 ?q))))
   :effect (and
     (not (at ?a ?p)) (at ?a ?q)
     (not (next-move ?a ?p ?q))
     (next-choose)))
```

The actions `choose-announce` and `announce` can be defined similarly. Announcing works by making all worlds where the agent has a different goal than its current position distinguishable to any other world for all agents.

E.g., our example instance from Figure 1 can then be defined using the following initial state and goal descriptions:

```
(:objects a1 a2 - agt l m r b - pos)
(:init (adj l m) (adj m l) (adj m r) ; ...
       (ind a1 w1 w2) (ind a1 w2 w1) ; ...
       (ind a2 w1 w3) (ind a2 w3 w1) ; ...
       (goal w1 a1 r) (goal w1 a2 l)
       ; ... (goals for w2, w3, w4)
       (des w1)  (next-choose))
(:goal (forall (?w - wld ?a - agt ?p - pos)
        (imply (and (des ?w) (goal ?w ?a ?p))
               (at ?a ?p))))
```

We tested our MAPF/DU planning domain using the myND planner of Mattmüller *et al.* (2010), which is to the

| Experiment | time |
|---|---|
| 2 agents, 4 cells, and 4 worlds | 0.55s |
| 3 agents, 6 cells, and 8 worlds | 11.5s |

Table 1: Case study.

best of our knowledge the only publicly available FOND planner that supports both strong acyclic plans as well as conditional effects. It also supports axioms, although we did not need them for our example. Table 1 shows the performance of the planner on the example instance from Figure 1 as well as on a slightly bigger version with three agents.

## Conclusion

In our paper, we have shown a decidable fragment of strong epistemic planning that has the same complexity than strong planning in FOND. We have also demonstrated how FOND planning can be used to generate subjectively strong plans. For future work, it is worth noticing that DEL can be used for modeling games. In particular, there is a translation from the game description language GDL-III to DEL (Engesser *et al.* 2018). There are some very interesting games which fall within our decidable fragment, one of which is Hanabi, which has gained some attention recently (Bard *et al.* 2019). While using a FOND planner does not seem to be feasible for problems of this size, it will be interesting to investigate how the idea of simulating perspective taking via nondeterminism can be incorporated into techniques such as Monte Carlo tree search or model-based reinforcement learning (e.g., value iteration in fully-observable MDPs).

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
