# OpenReview forum: "Planning for Implicit Coordination using FOND"
_icaps-conference.org/ICAPS/2019/Workshop/KEPS — KEPS 2019_

### Official Review · AnonReviewer1 · 2019-04-26
**Compilation to FOND**

**Rating:** 3
**Confidence:** 1

**Review:**

This paper defines a decidable fragment of epistemic planning that can be compiled to fully-observable nondeterministic (FOND) planning, and keeps the same computational complexity. A small case-study is provided to demonstrate the fruitfulness of the compilation.

Generally, the paper provides the necessary context to the user, and it is well contextualised. The theoretical side is sound, and the compilation approach is described at an appropriate level of detail. The example helps to understand part of the details, and gives some useful descriptions of the resulting operators.

As a suggestion for further improvements, and given the topic of the workshop, it would have been nice to provide some additional insights into the characteristics of the generated models. How can they be made easier to handle by planning engines? what are the most challenging aspects? Are there aspects of the language that are forcing the use of some caveats?

* Minor issues
- don't -> do not
- "As explained in Section, " -> the AAAI template does not enumerate sections.

---

### Official Review · AnonReviewer2 · 2019-05-08
**Compiling a Fragment of Epistemic Planning to FOND**

**Rating:** 3
**Confidence:** 2

**Review:**

The paper introduces a compilation scheme from a fragment of epistemic planning into Fully Observable Non-Deterministic (FOND) planning. It is shown (although without a formal proof) that the considered fragment of epistemic planning is as hard as FOND planning (EXPTIME-complete).

The approach is evaluated in two small instances of MAPF/DU (Multi-Agent Path Finding with Destination Uncertainty) domain. Even looking at these small instances, it can be seen that the approach does not scale well (which is not surprising given the large computational complexity). On the other hand, more instances/domains could shed more light into what we can expect from the approach.

The paper is not a typical Knowledge Engineering paper, although it concerns remodeling from one formalism to another which can, in my opinion, attract some attention at the workshop.


grammar: don't -> do not; doesn't -> does not